# Genome-Wide Differential DNA Methylation and miRNA Expression Profiling Reveals Epigenetic Regulatory Mechanisms Underlying Nitrogen-Limitation-Triggered Adaptation and Use Efficiency Enhancement in Allotetraploid Rapeseed

**DOI:** 10.3390/ijms21228453

**Published:** 2020-11-10

**Authors:** Ying-peng Hua, Ting Zhou, Jin-yong Huang, Cai-peng Yue, Hai-xing Song, Chun-yun Guan, Zhen-hua Zhang

**Affiliations:** 1School of Agricultural Sciences, Zhengzhou University, Zhengzhou 450001, China; yingpenghua@zzu.edu.cn (Y.-p.H.); zhoutt@zzu.edu.cn (T.Z.); jinyhuang@zzu.edu.cn (J.-y.H.); yuecaipeng@zzu.edu.cn (C.-p.Y.); 2Southern Regional Collaborative Innovation Center for Grain and Oil Crops in China, College of Resources and Environmental Sciences, Hunan Agricultural University, Changsha 430128, China; hyp19890413@163.com; 3National Center of Oilseed Crop Improvement, Hunan Branch, Changsha 430128, China; yingpenghua89@126.com

**Keywords:** allotetraploid rapeseed, DNA methylation, microRNA, nitrogen, nitrogen limitation adaptation, nitrogen use efficiency

## Abstract

Improving crop nitrogen (N) limitation adaptation (NLA) is a core approach to enhance N use efficiency (NUE) and reduce N fertilizer application. Rapeseed has a high demand for N nutrients for optimal plant growth and seed production, but it exhibits low NUE. Epigenetic modification, such as DNA methylation and modification from small RNAs, is key to plant adaptive responses to various stresses. However, epigenetic regulatory mechanisms underlying NLA and NUE remain elusive in allotetraploid *B. napus*. In this study, we identified overaccumulated carbohydrate, and improved primary and lateral roots in rapeseed plants under N limitation, which resulted in decreased plant nitrate concentrations, enhanced root-to-shoot N translocation, and increased NUE. Transcriptomics and RT-qPCR assays revealed that N limitation induced the expression of *NRT1.1*, *NRT1.5*, *NRT1.7*, *NRT2.1/NAR2.1*, and *Gln1;1*, and repressed the transcriptional levels of *CLCa*, *NRT1.8*, and *NIA1*. High-resolution whole genome bisulfite sequencing characterized 5094 differentially methylated genes involving ubiquitin-mediated proteolysis, N recycling, and phytohormone metabolism under N limitation. Hypermethylation/hypomethylation in promoter regions or gene bodies of some key N-metabolism genes might be involved in their transcriptional regulation by N limitation. Genome-wide miRNA sequencing identified 224 N limitation-responsive differentially expressed miRNAs regulating leaf development, amino acid metabolism, and plant hormone signal transduction. Furthermore, degradome sequencing and RT-qPCR assays revealed the miR827-NLA pathway regulating limited N-induced leaf senescence as well as the miR171-*SCL6* and miR160-*ARF17* pathways regulating root growth under N deficiency. Our study provides a comprehensive insight into the epigenetic regulatory mechanisms underlying rapeseed NLA, and it will be helpful for genetic engineering of NUE in crop species through epigenetic modification of some N metabolism-associated genes.

## 1. Introduction

Nitrogen (N) is a key macronutrient for plant growth, development, and yield capacity [1]. Nitrate (NO_3_^−^) is an N nutrient source that is preferentially absorbed by terrestrial plants [2]. Low nitrate availability is often observed in most agricultural soils; therefore, a huge amount of energy is consumed to annually apply large numbers of chemical N fertilizers to ensure crop production [3]. However, owing to leaching loss, soil surface runoff, denitrification, volatilization, and microbial consumption, more than half of the N nutrients are wasted [4]. Thus, increasing plant adaptability to N limitation is key to reducing N fertilizer application in modern agriculture production, which contributes to development of environmentally friendly agriculture and a sustainable ecosystem.

Epigenetic DNA modification, not changing genomic sequences, plays a pivotal role in maintaining genome stability, orchestrating gene expression across plant development, and regulating plant responses to environmental stimulations [5]. Recent studies have highlighted three epigenetic hallmarks: DNA methylation, histone modifications, and modification from small RNAs [6]. In plants, 5-methylcytosine (5 mC) is a major epigenetic signature that is modified by DNA methyltransferases. DNA methylation occurs not only predominantly at symmetric CG sites but also at CHG and asymmetric CHH regions (where H represents A, T, or C) [7]. Nutrient stresses in plants, such as N limitation [8], phosphate starvation [9,10], zinc deficiency [11], and sulfur deficiency [12], have been shown to change global DNA methylation at the whole genome level.

Endogenous microRNAs (miRNAs) are a class of evolutionarily conserved small noncoding RNA molecules that regulate plant transcriptional and post-transcriptional responses to various biotic and abiotic stresses [13]. Some miRNAs modulate plant adaptation to N limitation through modulation of their target genes [14]. *Arabidopsis* miR169 was downregulated under N deficiency, which induced an accumulation of its targets encoding nuclear factor Y subunit A family members. Plants overexpressing miR169 accumulated less N and showed greater sensitivity to N limitation, which was associated with the downregulation of important nitrate transporter (NRT) genes *AtNRT2.1* and *AtNRT1.1* [15]. In addition, the miR167/*ARF8* and miR393/*AFB3* regulatory circuits were shown to function in N-responsive regulatory networks, which regulated lateral root outgrowth [16]. Repression of an E3 ubiquitin ligase, NLA (Nitrogen Limitation Adaptation), by miRNA827 has been verified at the post-transcriptional (cleavage of mRNA) and translational levels under N deficiency [17]. N starvation triggered a striking decrease in miR164 levels with an increase in the mRNA level of its target, *ORE1* [18].

Allotetraploid *Brassica napus* (A_n_A_n_C_n_C_n_, 2*n* = 4*x* = 38) is an important oleaginous crop worldwide [19]. *B. napus* is more susceptible to N deficiency than grain crops to maintain optimal growth and development. However, the N use efficiency (NUE) of *B. napus* is much lower than those of other major crops [20]. Numerous duplicated segments and homoeologous regions are found within the *B. napus* genome [19], indicating that there might be complicated molecular regulatory networks involving NLA and NUE. To understand and explore epigenetic regulatory mechanisms underlying the adaptive responses of rapeseed plants to N limitation, the aim of our present study is to (i) characterize physiological responses, including carbon–N metabolism, root system architecture (RSA), and ionomic profiling, of rapeseed plants under N limitation, (ii) analyze differentially expressed miRNAs between N sufficiency and N limitation, as well as their targets genes, and (iii) identify differentially methylated regions/genes between N sufficiency and N limitation. Our study will provide helpful information for genetic engineering of NUE and NLA in plants through epigenetic modification of some N-metabolism-associated genes.

## 2. Results 

### 2.1. Physiological Adaptive Responses of Rapeseed Plants to N Limitation

Under N limitation, rapeseed plants showed smaller and chlorotic leaves (Figure 1A–D), which was also confirmed by lower biomasses (Figure 1B). Compared with the decrease in root (R) biomasses, a higher degree of reduction in the shoot (S) biomasses was observed under low N condition (Figure 1B). Based on the above result, we observed that N limitation led to an increase in the R/S ratio (Figure 1C). Moreover, limited N, compared with N sufficiency, resulted in over 75% reduction in the total leaf areas (Figure 1D). Flow of organic acids into amino acids was significantly decreased in plant shoots under N limitation [21]. Under low N, more organic acids were diverted to sugar compounds in the leaves (Figure 1E–G). Anthocyanin biosynthesis is usually considered as an adaptive response to N limitation [21], and we found that the anthocyanin concentration under N limitation was up to almost twofold of that under N sufficiency (Figure 1E,F). Moreover, the concentrations of some other carbohydrates, such as glucose, fructose, and sucrose, were also significantly increased under low N (Figure 1G). 

Besides great alteration in the shoot performance and metabolism (Figure 1A–G), N limitation also induced a dramatic change in rapeseed RSA. Although the root biomasses were markedly reduced (Figure 1B), total root length, root surface area, and root volume were significantly increased under N limitation (Figure 1H,I). However, root diameters were decreased by more than 40% under low N (Figure 1I), which might be a key factor causing reduction in root biomasses.

Subsequently, we examined the effect of N limitation on ionomic profiling in rapeseed plants (Figure 2A). Indeed, low N reduced nitrate concentrations in both shoots and roots (Figure 2A). However, compared with N sufficiency, a larger proportion of nitrate was translocated from roots to shoots (Figure 2B), which finally resulted in a significantly higher NUE under N limitation (Figure 2C). Besides the reduction in nitrate concentrations (Figure 2A), low N induced remarkable decreases in the concentrations of most cations, such as K^+^, Ca^2+^, Mg^2+^, and Cu^2+^, in both shoots and roots (Figure 2A). Among them, with exception, the concentrations of Fe^2+^ and Mn^2+^ in the shoots and Zn^2+^ in the roots were not significantly changed between high N and low N conditions (Figure 2A).

### 2.2. Transcriptional Adaptive Responses of Rapeseed Plants to N Limitation

In our previous study [22], we performed a genome-wide transcriptional analysis of rapeseed plants under sufficient N and limited N conditions. We identified a total of 4346 differentially expressed genes (DEGs) in the shoots and 9693 in the roots between the two N supply conditions, respectively [22]. In this study, to better understand molecular responses of rapeseed plants to N limitation, we used the online Panther program to retrieve the functional ontology of identified DEGs. The result showed that transporters (mainly nutrient transporters) and transferases (including glycosyltransferases and methyltransferases) were highly accumulated in both shoots and roots (Figure 2D). This indicated that nutrient transport and epigenetic modification, such as DNA methylation, might play crucial roles in the adaptive responses of rapeseed plants to N limitation stress.

To examine molecular responses of N-metabolism-associated genes to N deficiency, we determined their differential expression between high N and low N conditions by RT-qPCR assays (Figure 2E). In terms of N transporter genes that are responsible for N absorption, we focused on the dual-affinity nitrate transporter (NRT) gene, *NRT1.1/NPF6.3* [23], and the two-component high-affinity nitrate transport system genes, *NRT2.1* and *NAR2.1/NRT3.1* [24]. Obviously, the expression of three-type *NRTs* above-mentioned was significantly induced in the roots after exposure to 72-h N starvation. Furthermore, we investigated the expression of *NRTs* that are involved in long-distance translocation of N nutrients, such as *NRT1.5/NPF7.3* and *NRT1.8/NPF7.2*, which are responsible for root nitrate xylem loading and unloading [25,26], respectively. The results showed that N depletion elevated the mRNA level of *NRT1.5*, whereas repressed the *NRT1.8* expression. The expression of *NRT1.7/NPF2.13*, which facilitates N nutrient remobilization from source leaves to sinks [27], was also enhanced under N limitation. Moreover, the expression of a vacuolar nitrate influx transporter gene, *CLCa*, was lower under N limitation than under N sufficiency. Furthermore, N limitation repressed the expression of the nitrate reductase (NIA/NR) gene *NIA1* and increased the mRNA level of the glutamine synthetase (Gln/GS) gene *Gln1;1/GS1;1*. 

Considering that DNA methyltransferase terms were highly enriched in the GO analysis, we decided to investigate differential expression profiling of DNA methyltransferase genes. We identified a total of 12 DNA methyltransferase genes that were differentially expressed in the shoots and roots under N limitation (Figure 2F). Among the eight DEGs in the shoots, most of them were upregulated, and a similar expression pattern was also observed in the roots (Figure 2F). Subsequently, we selected a DNA methyltransferase gene (BnaC06g23520D) and examined its expression using RT-qPCR assays; the result confirmed that N limitation led to a significant increase in the DNA methyltransferase expression (Figure 2G). The finding indicated that N limitation might change DNA methylation status in rapeseed plants. 

### 2.3. Genome-Wide High-Resolution DNA Methylation Fingerprints in Response to N Limitation

To further identify the effect of N limitation on genome-wide DNA methylation, we conducted a global comparison of DNA methylation patterns between N sufficiency and N limitation by whole genome bisulfite sequencing (WGBS). The WGBS generated a total of 120 million raw reads for sample library by paired-end sequencing, and more than 90% of them were clean reads, which covered more than 30× depth of the rapeseed genome (Appendix A). More than 75% cytosines were covered by at least five sequencing reads in the rapeseed genome. Low base calling error rates (Q_20_ < 5%, Q_30_ < 10%) and high bisulfite conversion rates (> 99%) suggested high quality of the global WGBS data (Appendix A).

In general, the genome-wide DNA methylation level under the CG context was the highest, followed by that under CHG and CHH contexts (Figure 3A, Appendix A). The tendency of variations in DNA methylation across genomic regions was consistent in the shoots and roots under both N sufficiency and N limitation (Appendix A). The average methylation levels were uneven among the 19 chromosomes, and the methylation abundance of the A_n_ subgenome was significantly lower than that of the C_n_ subgenome from all the DNA libraries under all the CG, CHH, and CHG contexts (Figure 3B, Appendix A). To better understand the relationship between DNA methylation and gene expression, we divided the rapeseed genome into several functional categories, including gene promoter, gene body, and 2.0-kb downstream regions of genes. The methylation degrees of gene promoters and downstream regions were obviously higher than those of gene bodies (Appendix A), with a slightly higher methylation level for introns than for exons (Appendix A). 

To further identify differential DNA methylation profiling between N sufficiency and N limitation, we characterized genome-wide differentially methylated regions (DMRs) in the shoots and roots. From the global view, there were 4474 and 4485 differentially methylated cytosines for CG in the shoots and roots, 3205 and 1200 for CHG, and 18,987 and 4728 for CHH, respectively (Figure 3C–E, Appendix A). In detail, there were high peaks of DMR density on the A2/A9 chromosomes of the A_n_ subgenome and on the C2/C4/C7 chromosomes of the C_n_ subgenome (Figure 3C–E, Appendix A). The majority of DMR lengths under the CG and CHG contexts were fewer than 100 nucleotides (nts), which were obviously shorter than those under the CHH context (Appendix A). Different genomic features of differential methylation were observed between N sufficiency and N limitation (Figure 3F–H and Figure 4, Appendix A). In the CG context, we did not identify significant methylation differences within all the gene body and their 2.0-kb upstream (namely promoter) and downstream regions. In the CHG context, numerous hypermethylation was detected under N limitation compared with that under N sufficiency. However, in the CHH context, a discrepancy in response to N limitation was observed between the shoots and roots; in the shoots, N limitation led to DNA hypomethylation, but the pattern was opposite in the roots. 

In total, we identified 2389 differentially methylated genes (DMGs), including 1206 hypermethylated genes and 1284 hypomethylated genes, in the shoots between N sufficiency and N limitation. In the roots, we detected 2705 DMGs, including 1511 hypermethylated genes and 1316 hypomethylated genes between N sufficiency and N limitation (Appendix A). The majority of DMGs were identified in the CG genomic context, the number of which was subsequently followed by DMGs in the CHH and CHG regions (Appendix A). 

The GO of DMGs and their involvement in KEGG pathways were used to identify the function of DMGs between N sufficiency and N limitation. Cellular and metabolic processes belonging to the biological process item, cell parts and protein complexes belonging to the cellular component group, and transcription binding and transporter activities belonging to the molecular function category were the most overrepresented in the GO analysis (Figure 5A). Furthermore, we mapped the DMGs onto KEGG pathways in which they were involved (Figure 5B). In the shoots, the items of starch, sucrose, and anthocyanin (phenylpropanoid pathway) metabolism, photosynthesis, and protein processing in the endoplasmic reticulum were highly enriched. This result indicated that N limitation might have a significant effect on the methylation of carbon–N-metabolism-associated genes. In addition, the pathways involving transcription-related pyrimidine and RNA polymerase were also overaccumulated. In the roots, the terms of protein export, ubiquitin-mediated proteolysis, and proteasome and amino acid (glutamate, etc.) metabolism were largely enriched. It suggested that DNA methylation might occur on the genes involved in organic N-derived N recycling. 

Therefore, we focused on the effect of N limitation on DNA methylation of some N-metabolism associated genes. Among the genome-wide *NRT1.1*/*NPF6.3* homologs and nine high-affinity *NRT2.1* homologs in the rapeseed genome [28], we found only a CHH hypomethylation site located in the promoter of an *NRT1.1* homolog (BnaC08g15370D) and a CHH hypomethylation site located in the promoter region of an *NRT2.1* homolog (BnaC06g04560D) in the roots under N limitation, respectively (Table 1). N starvation induced an increase in the expression of all the *NRT1.1/NPF6.3* and *NRT2.1* homologs (Figure 2E), which might be correlated with their hypomethylation in their promoter regions. However, we also found that DNA methylation in promoter regions did not consistently repress gene expression levels. For instance, we identified a hypermethylated site in the promoter of *NRT1.5/NPF7.3* (BnaC05g24580D) in the roots and a hypermethylated site in the promoter region of *NRT1.7/NPF2.13* (BnaC06g30920D) in the shoots of rapeseed plants under N limitation (Table 1), whereas their expression was induced by N limitation (Figure 2E). Furthermore, some other *NPF* family genes, such as *NRT1.6*/*NPF2.12*, *NRT1.9/NPF2.9*, *NRT1.11/NPF1.2* and *NAXT1 (nitrate excretion transporter 1)/NPF2.7*, also showed significantly differential methylation between N sufficiency and N limitation (Table 1). *CLCa*, a chloride channel gene, is mainly responsible for vacuolar nitrate influx [29], and we found a hypermethylated site in its promoter region (Table 1), which was consistent with its reduced expression under N limitation (Figure 2E).

In addition to the molecular identification of *NRTs*, we also tested a hypermethylated site in an intron of *Nitrate Reductase 1* (*NIA1*, BnaA02g18610D) and a hypermethylated site in an exon of *Glutamate Synthetase 1;1* (*Gln1;1*, BnaC04g30250D) in the roots under N limitation (Table 1), which might be associated with their enhanced transcriptional expression [22]. These above findings indicated that N limitation had a significant effect on N-metabolism-associated genes, including those responsible for N absorption, translocation, remobilization, and assimilation. 

### 2.4. Transcriptional Regulation of Rapeseed miRNAs in Response to N Limitation

To determine molecular responses of miRNAs to N limitation and identify the genes targeted by differentially expressed miRNAs, we constructed a total of 12 small RNA libraries from N-sufficient shoots (shoot_0 h), N-sufficient roots (root_0 h), N-deficient shoots (shoot_72 h), and N-deficient roots (root_72 h) for high-throughput sequencing of small RNAs. After removing low-quality reads and adapter sequences, we obtained about 16 million clean reads, corresponding to ~0.87 Gb of sequencing data, for each sample (Appendix A). Based on the results of normalized expression between each two biological replicates, the Pearson correlation coefficients (*R*) were calculated, most of which were more than 0.92 (Appendix A). Low base calling rates, similar GC content, and high correlation among different biological replicates indicated high quality of the sequencing data. 

Length distribution of miRNAs frequently reflects the specificity of a specific species or tissue [30]. We found that the nucleotide (nt) lengths of Bna-miRNAs ranged from 18 nts to 30 nts, and the miRNAs with 24 nts were most abundant in all the miRNA libraries (Figure 6A). Distribution characteristics of the miRNA nucleotide lengths were similar to those of previous reports in *B. napus* [31,32]. Most Bna-miRNA sequences showed a strong bias for a uridine (U) at the first nucleotide position (Appendix A). Before the N-limitation treatment, namely at 0 h (N sufficiency), most of the 24-nt long miRNAs had an apparent preference for U in both shoots and roots; however, at 72 h (N limitation), they had a bias for guanine (G) (Appendix A). The hierarchical clustering and heat maps of genome-wide miRNA expression revealed that similar expression patterns occurred in the three biological replicates of each sample. However, significantly differential expression profiling of miRNAs was identified among different plant tissues or N treatments (Appendix A). 

In this study, the miRNAs showing high expression levels, accounting for a major proportion of the genome-wide miRNAs of rapeseed (Appendix A), were easily detected in the subsequent studies. We found that different members in the same miRNA family exhibited similar expression levels, such as miR164s and miR167s in the shoots and miR156s and miR171s in the roots. A total of 123 and 101 differentially expressed miRNAs in the shoots and roots, respectively, were identified, more than half of which were downregulated by low N (Figure 6B). Furthermore, through a Venn diagram analysis, we found that 20 and 42 differentially expressed miRNAs were upregulated and downregulated in both shoots and roots, respectively, (Figure 6C). To identify biological function of the genes targeted by differentially expressed miRNAs in *B. napus* during exposure to N deficiency, we plotted GO enrichment and KEGG pathways in which they were involved. We found that most potential targets of the conserved miRNAs were transcription factor genes, such as *MYBs* and *Auxin-responsive factors* (*ARFs*). For example, both miR160 and miR167 target ARF transcription factors [33,34], and miR156 and miR162 target squamosal promoter-binding-like (SPL) proteins and Dicer-liker (DCL) proteins. In addition, target genes that are implicated in leaf development and cell division, as well as phytohormone signaling, were also highly accumulated (Figure 6D). In our previous study, we identified numerous *MYB* transcription factor genes that were differentially expressed between N sufficiency and N limitation [22], which might be mainly involved in the anthocyanin biosynthesis in response to N limitation. For the KEGG analysis, the pathways involving environmental adaptation, carbon (such as carbohydrate and lipid) and N (such as amino acid) metabolism, and membrane transport were significantly accumulated (Figure 6E).

In general, miRNAs are highly conserved among different plant species [35]. We used the miRbase database [36] to detect known miRNAs that were differentially responsive to high N and low N conditions. A total of 34 known differentially expressed miRNAs belonging to 22 families and 89 novel were identified in the shoots (Figure 7A,B). We found a total of 22 known differentially expressed miRNAs belonging to 16 families and 79 novel differentially expressed miRNAs in the roots (Figure 7C,D). Among the known miRNAs, the Bna-miR169/Bna-miR167 families in the shoots and the Bna-miR156/Bna-miR171 families in the roots had the most miRNA members, and the Bna-miR171 family members presented the largest differential expression between N sufficiency and N limitation in both shoots and roots (Figure 7A,C).

In the shoots, both Bna-miR164a and Bna-miR164b, which regulated leaf senescence, were significantly induced under N limitation (Figure 7A). Bna-miR156, which regulated production of anthocyanin, was downregulated in the shoots and was upregulated in the roots [37]. In the roots, both Bna-miR160a and Bna-miR171 family members (including miR171a, miR171f, and miR171g), which regulated root growth [38], were significantly elevated under N limitation compared with those under N sufficiency (Figure 7C), and they might facilitate N-starvation-induced root elongation. Overexpression of miR169 repressed the expression of the major *NRT* genes *AtNRT1.1* and *AtNRT2.1* [15]. In this study, the expression of Bna-mi169m was decreased in the roots under N limitation (Figure 7C), which might be associated with the enhanced transcript levels of *BnaNRT1.1s* and *BnaNRT2.1s* (Figure 2E). Furthermore, downregulation of miR164 combined with the upregulation of NAC1 produced more lateral roots [39]. However, we found that two miR164 homologs (Bna-miR164a/b) were upregulated in the shoots and roots under N limitation (Figure 7A,C). Therefore, we presumed that Bna-miR164a/b was not involved in the N-starvation-induced lateral root proliferation.

### 2.5. Degradome Sequencing-Assisted Identification of Genes Targeted by Differentially Expressed miRNAs in Response to N Limitation

To identify miRNA-directed mRNA cleavage events in rapeseed plants under N limitation, we subjected rapeseed plants under high N and low N to a high-throughput degradome sequencing platform. A total of 18,370,535 clean reads of 23.67 million raw reads in the shoots and 20,131,661 clean reads of 20.23 million raw reads in the roots were obtained, and more than 70% of the clean reads were matched with the reference sequences assembled from the rapeseed transcriptome database (Appendix A). In total, we identified 31 genes identified as targets of known miRNAs, which covered 30 miRNA families, consisting of 48 members (Appendix A).

Under N limitation, we found not only chlorophyll decrease, anthocyanin accumulation, leaf branch reduction, and leaf senescence in the shoots but also induction of primary and lateral roots (Figure 1), which were considered as adaptive nutrient morphogenesis responses of rapeseed plants to N limitation [22]. To identify the miRNA-mediated regulatory mechanisms underlying these morpho-physiological responses, we tested the expression of several related miRNAs and their target genes. In our present and previous studies, we identified increased expression of Bna-miR827 in the shoots (Figure 7A). The involvement of the At-miR827-AtNLA-AtORE1 cascade pathway in the N-limitation-induced leaf senescence in *Arabidopsis* was validated [17]. We found a higher transcript accumulation of Bna-miR827 in the leaves of rapeseed plants under N limitation than that under N sufficiency (Figure 7A), which repressed the expression of *NLA* [22] and further led to the enhanced expression of ORE1, a major transcription factor regulating leaf senescence.

In *Arabidopsis*, *SCL6 (SCARECROW-Like 6)*, which is targeted by miR171, plays an important role in the regulation of shoot branch production and leaf morphogenesis [40]. In this study, we found that the identified Bna-miR171f was induced in both shoots and roots under N limitation (Figure 7A,C). The mature sequence of Bna-miR171f was mapped onto the 3’-end of its precursor (Figure 8A), and it was highly identical with the corresponding homolog in *Arabidopsis* (Figure 8B). Stem–loop RT-qPCR confirmed that the expression of Bna-miR171f was induced by N limitation in both shoots and roots (Figure 8C). Furthermore, we examined the expression of the *SCL6* homologs in *B. napus* and found that only *BnaAn.SCL6b* (BnaAnn18550D) was significantly repressed by N limitation (Figure 8D). The antagonistic expression between Bna-miR171f and *BnaAn.SCL6b* indicated their potential interaction. Subsequently, we used degradome sequencing to further validate the presumption. The degradome result showed that the Bna-miR171f-mediated cleavage site fully matched the transcript of *BnaAn.SCL6b*, whose transcript had only an obvious peak showing the cleavage site (Figure 8E,F).

In *A. thaliana*, overexpression of *ARF17*, which is targeted by miR160, leads to root length reduction [41]. In this study, we identified that Bna-miR160a, induced by N limitation (Figure 7A), was mapped onto the 5’-end of its precursor (Figure 9A), and that it shared a high identity with its homolog in *Arabidopsis* (Figure 9B). RT-qPCR confirmed that the expression of Bna-miR160a was induced by N limitation in both shoots and roots (Figure 9C). Among the *ARF17* homologs in *B. napus*, only *BnaC6.ARF17a* (BnaC06g20640D) was differentially expressed between high N and low N conditions (Figure 9D). Furthermore, the degradome data showed that the Bna-miR160a-mediated cleavage site fully matched the transcript of *BnaC6.ARF17a*, whose transcript had only an obvious peak showing the cleavage site (Figure 9E,F). 

## 3. Discussion

N nutrients play pivotal roles in modulating crop growth, organ development, and seed productivity [42]. Currently, great process has been made in uncovering the genetic and molecular mechanisms underlying NUE and NLA in the model plants *Arabidopsis* and rice [43,44,45,46,47,48,49]. In addition to characterization of the aforementioned functional structural genes, epigenetic modification, such as phosphorylation of *NRT1.1* [50], methylation of *NRT2.1* [51], and post-transcriptional cleavage of NLA by miR827 [17], is also significant for the regulation of NUE and NLA in plants.

Rapeseed has a higher demand for N nutrients than do many other crop species [20,52]. For the purpose of guaranteeing seed yield production, enhancing rapeseed NLA is a core route for not only reducing N fertilizer applications but also increasing NUE. However, the implicated mechanisms underlying NUE and NLA in allotetraploid rapeseed remain unclear. In our study and other previous studies, genome-wide transcriptomic and proteomic responses of rapeseed plants to N limitation were investigated through high-throughput sequencing [5,22]. Furthermore, in this study, we employed WGBS and small RNA sequencing to identify epigenetic modification, including DNA methylation and miRNA cleavage, in regulating adaptive responses of rapeseed plants to N limitation.

### 3.1. N Limitation Triggered Adaptation and NUE Enhancement

Nitrate functions not only as a N nutrient source, but as a signal molecule affecting plant growth and development in plants by inducing expression of auxin-related genes [53], modulating leaf growth and development [54,55], and regulating root system architecture [56,57,58,59,60]. An *Arabidopsis* mutant, *nla*, shows leaf early senescence and does not accumulate anthocyanin under N limitation, not exhibiting an adaptive response of plants to limited N nutrients [17]. Under N limitation, the rapeseed leaves and hypocotyls rapidly accumulated large quantities of anthocyanins (Figure 1A). Anthocyanins belong to important flavonoid categories, and they can protect plants from N-deficiency-induced damages of reactive oxygen species [61]. Under N limitation, the NUE of rapeseed plants was significantly increased (Figure 2C). Startup of limited N induces leaf senescence, which triggers N nutrient remobilization from senescent leaves to developing leaves or other organs, such as seeds [17]. However, weak NLA induces leaf early senescence, easily resulting in leaf dropping and detaching from plants before that N can be sufficiently remobilized to the sink organs [17]. Therefore, it suggested that improving plant adaptation to N limitation is an urgent route for NUE enhancement.

As a major determinant of nutrient uptake, developmental plasticity of RSA has a significant impact on plant growth performance and nutrient use efficiency in nutrient-depleted soils [62]. RSA-based genetic improvement is proposed to be a feasible tool for the rapeseed NUE enhancement [63]. In turn, RSA is also influenced by N availability, which allows plants to optimize N acquisition from growth medium by modulating RSA [64]. In this study, we found that under N deficiency (0.30 mM NO_3_^−^), the primary roots of rapeseed became longer, combined with more lateral roots than those under N sufficiency (6.0 mM NO_3_^−^) (Figure 1H). It might contribute to root foraging for N nutrients in the growth media (Figure 10). Based on these findings, we proposed that identification and characterization of the genes regulating RSA is urgent for the genetic improvement of plant NLA and NUE. 

### 3.2. Genome-Wide Differential DNA Methylation Reveals Pivotal Roles of Epigenetic Regulation in Adaptive Responses of Rapeseed to N Limitation

DNA methylation plays an important role in regulation of transposon silencing, genome stability, and gene expression in plants, and its pattern is affected by environmental biotic and abiotic factors. Nutrient stresses in soils, such as phosphate starvation and N deficiency, were shown to alter DNA methylation status within plants [8,65]. Prior to this study, few research studies were conducted to identify and characterize the DNA methylation fingerprints of allotetraploid rapeseed under N limitation. Therefore, to illustrate the effect of N limitation on DNA methylation, we showed how rapeseed plants remodeled methylation landscapes of genomic DNA across gene structures in response to N limitation.

Despite derivation from a common ancestor, the diploid *Brassica oleracea* (C_o_C_o_) and *Brassica rapa* (A_r_A_r_) contain 38.80% and 21.47% transposable elements (TEs), respectively, in their assembled genomes, respectively [66]. The above facts lead to an asymmetric distribution of TEs in the A_n_ and C_n_ subgenomes of the assembled *B. napus* genome (A_n_A_n_C_n_C_n_), which may lead to different methylation patterns between the two subgenomes [19]. Regulatory roles of DNA methylation in gene transcription are dependent on plant tissue types, methylated genomic contexts, and target gene regions [67]. DNA methylation of regulatory elements, such as gene promoters and enhancers, potentially repress gene expression by altering chromatin structures and inhibiting transcription initiation [68]. In this study, we identified hypomethylation in the gene promoter regions of rapeseed *NRT1.1* and *NRT2.1* (Table 1), which might be involved in their transcription activation under N limitation (Figure 2E). A hypermethylated site in an exon of *Gln1;1/GS1;1* (BnaC04g30250D) was identified in the roots of rapeseed plants under N limitation (Table 1), whereas its expression was enhanced (Figure 2E). Therefore, gene body methylation does not necessarily reduce the gene expression [69], but instead plays a positive role in transcription activation in some plant species, such as *Arabidopsis* [70], rice, and maize [71]. Furthermore, we found that DNA methylation patterns of some genes were not consistent with their transcriptional expression, such as *NRT1.5* and *NRT1.7* (Figure 2E, Table 1). MYB59 directly bound to the *NRT1.5* promoter and positively regulated its expression [72]. NLA which encodes a RING E3 ubiquitin ligase, interacts directly with NRT1.7 in the plasma membrane and promotes NRT1.7 degradation [17]. The above results indicated that other regulatory mechanisms—for instance, transcription factor binding or post-translational modification—might be involved in the transcriptional modulation of these genes above-mentioned.

### 3.3. Global miRNA and Degradome Sequencing-Assisted Identification of Differentially Expressed miRNAs Uncover Epigenetic Regulatory Mechanisms Underlying N-Limitation-Induced Morphogenesis

To identify the roles of mRNA, miRNA, and their targets in regulating these N-deficiency-induced morphogenesis, we employed high-throughput genome-wide mRNA sequencing, miRNA sequencing, degradome sequencing, and RT-qPCR assays to identify differential expression profiling in response to N limitation. Genome-wide mRNA transcriptome sequencing and RT-qPCR assays revealed that under N limitation, the expression of *NRT1.1*, *NRT1.5*, *NRT1.7*, *NRT2.1*/*NAR2.1*, and *Gln1;1* was significantly increased, whereas the transcript levels of *CLCa*, *NRT1.8*, and *NIA1* were reduced (Figure 2E). This finding indicated that limited N improved plant NUE by enhancing efficient N absorption, reducing vacuolar N storage and root unloading, facilitating root-to-shoot long-distance translocation, improving source-to-sink remobilization, and strengthening N assimilation (Figure 10).

Auxin-mediated nitrate signaling by *NRT1.1* participates in adaptive response of *Arabidopsis* RSA to the spatial heterogeneity of nitrate availability [73]. Moreover, *NRT2.1* is also shown to be involved in nitrate-dependent root elongation by regulating auxin transport to roots [74]. Enhanced expression of nitrate transporters plays a key role in the regulation of RSA-mediated adaptive responses to N limitation through modulation of auxin signaling. Furthermore, we employed miRNA and degradome sequencing to identify key miRNAs and their targets that were involved in regulating adaptive responses of rapeseed plants to N limitation. In general, we found that the genes that are involved in leaf development and phytohormone (particularly auxin) signaling were enriched (Figure 6D,E). Therefore, we presumed that differentially expressed miRNAs might be implicated in the regulation of N-limitation-induced shoot and root morphogenesis, and we proposed that a genetic engineering approach by manipulation of miRNA expression may be a feasible route for the improvement of NUE and NLA in rapeseed plants.

## 4. Material and Methods

## 4.1. Plant Materials and Growth Conditions

*B. napus* seedlings (cultivar, Zhongshuang 11) were hydroponically grown using the Hoagland solution, which was constantly aerated throughout experiments and refreshed every 5 days. Basic nutrition solution contained 1.0 mM KH_2_PO_4_, 5.0 mM KNO_3_, 5.0 mM Ca(NO_3_)_2_·4H_2_O, 2.0 mM MgSO_4_·7H_2_O, 0.050 mM EDTA-Fe, 9.0 µM MnCl_2_·4H_2_O, 0.80 µM ZnSO_4_·7H_2_O, 0.30 µM CuSO_4_·5H_2_O, 0.37 µM Na_2_MoO_4_·2H_2_O, and 46 µM H_3_BO_3_. Uniform rapeseed seedlings were cultivated in an illuminated growth chamber (300–320 µmol m^−^^2^ s^−^^1^; 24 °C daytime/22 °C night; 16 h light/8 h dark) with relative humidity of 75%. For nitrate-depletion treatments, rapeseed seedlings after 5-day germination were first hydroponically grown under high nitrate (6.0 mM) for 10 days, and then the rapeseed plants were transferred to low nitrate (0.30 mM). At the times of 0 h and 72 h after N limitation treatment, rapeseed shoots and roots were individually sampled for isolation of genomic DNA and total RNA. The solution concentration of N nutrients was adjusted by reducing KNO_3_ and replacing Ca(NO_3_)_2_ with CaCl_2_, and K^+^ was complemented by adding KCl [63].

## 4.2. Quantification of Morpho-Physiological Characteristics

Total leaf areas of rapeseed plants were determined using an LI-COR LI-3100C leaf area meter. Anthocyanin was isolated using extraction buffer (CH_3_OH:H_2_O:H_2_O_2_ = 60:13:2, *v*/*v*/*v*); then, the purified extract was assayed by a UV-1800 spectrophotometer on the absorbance wavelengths at 530 and 657 nm [22]. The roots of rapeseed seedlings cultivated under hydroponic culture system were subjected to a image scanner, and then total root length, root volume, root surface area, and root average diameter were analyzed through WinRHIZO Pro (Regent Instruments, QC, Canada) [75]. Soluble sugars were isolated as described by [21], and were assayed for glucose, fructose, and sucrose concentrations using a commercially available kit (Megazyme, http://www.megazyme.com/).

Harvested rapeseed plants were oven-dried at 65 °C until a constant weight was obtained, and then the dried samples were digested in a HNO_3_/HClO_4_ (4:1, *v*/*v*) mixture at 200 °C until completely digested. Concentrations of mineral elements were quantified by ICP-MS (NexION^TM^ 350X; PerkinElmer, Massachusetts, USA). Fresh tissues of rapeseed plants were sampled and bathed in boiling water for 30 min; then, they were assayed for nitrate concentrations using a continuous flow autoanalyzer (AA3, Seal Analytical Inc., Southampton, UK) [76]. After treatment at 120 °C for 30 min, the samples were dried at 65 °C for 72 h, and then were milled to a fine powder. Total N concentrations of oven-dried plant tissues were determined with a carbon and N analyzer (Elemental Analyzer EA1108; Carlo Erba Strumentazione, Milan, Italy). NUE values were calculated as the following formula: NUE = total plant dry weight (g)/total plant N accumulation (g) [77].

## 4.3. DNA and RNA Isolation, Quantification, and Qualification

Genomic DNA (gDNA) and total RNA was isolated using 2% CTAB and TRIzol, respectively. A Nanophotometer^®^ spectrophotometer (IMPLEN, CA, USA) was used to check the purity of DNA and RNA, and their concentrations were measured using a Qubit^®^ DNA and RNA Assay Kit in a Qubit^®^ 2.0 Fluorometer (Life Technologies, CA, USA). An RNA Nano 6000 Assay Kit of an Agilent Bioanalyser 2000 system (Agilent Technologies, CA, USA) was used to assess RNA integrity by determining the RNA integrity number.

## 4.4. Library Preparation for Whole Genome Bisulfite Sequencing (WGBS) and miRNA Sequencing

A total amount of 5.0 µg gDNA spiked with λDNA was fragmented by sonication to 200–300 bp with a Covaris S220, which followed by end repair and adenylation. Cytosine-methylated barcodes were ligated to the sonicated DNA. The DNA fragments were treated twice with bisulfite using EZ DNA Methylation-GoldTM Kit (Zymo Research) before the ssDNA fragments were amplified by PCR using KAPA HiFi HotStart Uracil + ReadyMix (2X). Library concentration was quantified by Qubit^®^ 2.0 Fluorometer (Life Technologies, CA, USA) and quantitative PCR, and the insert size was assayed on an Agilent Bioanalyser 2100 system (Agilent technology).

A total of about 3.0 µg RNA per sample was used for construction of small RNA libraries, which was generated using NEBNext^®^ Multiplex Small RNA Library Prep Set for Illumina^®^ (NEB, USA). Briefly, the NEB small RNA adapters were ligated to the 3′ and 5′ ends of miRNAs. Reverse transcription primers were hybridized to the excess of 3′ small RNA adapter, and the first strand DNA was synthesized using the M-MuLV reverse transcriptase (RNase H^−^). PCR amplification was performed using LongAmp Taq 2X Master Mix, small RNA primers from Illumina, and index primers. PCR products were purified through 8% polyacrylamide gel electrophoresis, and the DNA fragments corresponding to 140–160 bp were recovered. Library quality was assessed on an Agilent Bioanalyser 2100 system (Agilent Technologies, CA, USA).

## 4.5. Clustering and Sequencing

For WGBS, the DNA libraries were sequenced on an Illumina Hiseq 4000, and 150 bp paired-end reads were generated.

For the miRNA sequencing, the clustering of index-code samples was performed on a cBot Cluster Generation system using TruSeq SR Kit v3-cBOt-HS (Illumina). After cluster generation, the libraries were sequenced on an Illumina Hiseq 2500 platform, which generated 50-bp single-end reads.

## 4.6. Data Analysis

### 4.6.1. Quality Control

Read sequences (FASTq) produced by the Illumina pipeline were preprocessed through Trimmomatic (v. 0.35) as follows: first, reads containing adapter sequences were filtered out; second, some reads having ambiguous bases in their sequences were removed; third, reads with more than 50% low quality bases (PHRED score ≤ 20) were excluded. Subsequently, Q_20_, Q_30_, and GC content of clean data were calculated for further analysis.

### 4.6.2. Reads Mapping to the Reference Sequences

Bismark (v. 0.12.5) was used to perform alignments of bisulfite-treated reads to the reference genome of *B. napus* (ZS11) annotated in the BnPIR (*Brassica napus* pan-genome information resource) (http://cbi.hzau.edu.cn/bnapus/index.php) database. The reference genome was first transformed into the bisulfite-converted version, and then were indexed using Bowtie2. Sequence reads were also transformed into the fully bisulfite-converted versions (C-to-T and G-to-A conversion) before they were aligned to the similarly converted versions of the genome in a directional manner. Sequence reads that produced a unique best alignment from the two alignment processes (original top and bottom strand) were then compared with the normal genomic sequences, and the methylation states of all cytosine positions in the read were inferred. Read pairs that shared the same coordinates in the rapeseed genome were regarded as duplicated and were removed before methylation state determination to avoid potential calculation bias of the methylation level. The sodium bisulfite non-conversion rate was calculated as the percentage of cytosine sequenced at cytosine reference positions in the lambda genome.

For small RNA sequencing data, clean reads were firstly obtained by removing contaminants, adapters, and low-quality reads. Remaining unique RNAs were mapped to the rapeseed reference genome (http://cbi.hzau.edu.cn/bnapus/index.php) using the SOAP2 program [78]. Sequences with a perfect match were retained for further analysis. Using Bowtie tools “soft”, the clean reads were aligned respectively with the Silva database (http://www.arb-silva.de/), the GtRNAdb database (http://lowelab.ucsc.edu/GtRNAdb/), the Rfam database (http://rfam.xfam.org/), and the Repbase database (http://www.girinst.org/repbase/) in order to filter out ribosomal RNA (rRNA), transfer RNA (tRNA), small nuclear RNA (snRNA), small nucleolar RNA (snoRNA), and other ncRNA and repeats.

### 4.6.3. miRNA Alignment and Target Genes Prediction

The miRBase 22.1 (http://www.mirbase.org/) [79] database was used to retrieve published miRNA sequences and functional annotation. miRDeep2 was used for the identification of novel and known miRNAs. The miREvo and miRDeep2 databases were integrated to predict novel miRNAs through exploring secondary structures, Dicer cleavage sites, and minimum free energy of small RNA tags without annotation. Prediction of target genes of miRNAs was performed by psRobot_tar in psRobot.

### 4.7. Differential Analysis of DNA Methylation and miRNA Expression

After removing low-quality reads (Q score < 28), 150 bp paired-end WGBS reads were mapped to the rapeseed genome using Bismark software. Only cytosines that were covered by at least three reads were considered and counted [80]. The methylation level of individual cytosines was calculated as the ratio of mC to total cytosines [mC/(mC+un-mC)]. Fold change at each site was calculated by comparing the methylation ratio under N deficiency to the methylation ratio under N sufficiency. An mC fold change greater than 1.2 was considered to indicate differential methylation. Densities and average methylation levels were calculated in 50 kb units. The Kruskal–Wallis rank sum test was performed on each bin for the mCs in CG, CHG, and CHH contexts to identify significant differences between N sufficiency and N deficiency. The *p*-value of the bins was adjusted using BH multiple test correction. Bins with corrected *p* < 0.05 were defined as differentially methylated regions (DMRs). Differentially methylated genes (DMGs) were defined as genes whose gene body or promoter region (upstream 2.0 kb from the transcription starting site) have at least 1 bp overlap with their DMRs. Hypermethylation/hypomethylation status was assigned to each DMR based on the differences in methylation levels between the two N levels. If the methylation level of a DMR under N sufficiency was lower than that under N limitation, the DMR was considered a hypermethylated site and vice versa [81].

miRNA expression levels were estimated by transcripts per million (TPM). Corrected *p* ≤ 0.05 and |log_2_(fold change)| ≥ 0.60 were set as the threshold for the identification of differentially expressed miRNAs using the DESeq R package (1.10.1).

### 4.8. Gene Ontology and KEGG Enrichment Analysis

Differentially expressed genes (cutoff: *p* < 0.05 and false discovery rate < 0.05) that were identified by transcriptome sequencing were submitted to the online Protein ANalysis THrough Evolutionary Relationships (PANTHER) classification system (http://www.pantherdb.org/) for Gene Ontology (GO) enrichment analysis [82].

In terms of the data of WGBS and miRNA sequencing, the GOseq R package based on Wallenius non-central hyper-geometric distribution was used to perform GO enrichment analysis of the target genes. DEG enrichment in Kyoto Encyclopedia of Genes and Genomes (KEGG) (http://www.kegg.jp/) pathways was assessed using the KOBAS 3.0 software (http://kobas.cbi.pku.edu.cn/kobas3) [83,84]. Both *p*-values and false discovery rate values less than 0.05 were used to identify overrepresentation of GO categories and KEGG pathways. 

### 4.9. Degradome Analysis

For degradome sequencing, the rapeseed seedlings, ZS11, which were first hydroponically grown under high nitrate (6.0 mM) for 7 days, were then transferred to low nitrate (0.30 mM). At the times of 0 h and 72 h after N limitation treatment, rapeseed shoots and roots were individually sampled for RNA isolation. Parallel analysis of RNA ends libraries was performed on an Illumina’s Cluster Station and sequencing was subsequently conducted on Illumina GAIIx. CLEAVELAND 3.0 was used for the sequencing data analysis.

### 4.10. Reverse Transcription Quantitative PCR (RT-qPCR) Assays

Shoots and roots of rapeseed plants were individually harvested and immediately stored at −80 °C until RNA isolation. Each sample contained three independent biological replicates for transcriptional analyses. Reverse transcription quantitative PCR (RT-qPCR) assays were used to determine the gene expression. After removing genomic DNA from the RNA samples with RNase-free DNase I, complementary DNA (cDNA) synthesis was performed using the PrimeScript^TM^ RT reagent Kit with gDNA Eraser (Perfect Real Time) (TaKaRa, Shiga, Japan) with total RNA as the templates. Quantitative analysis of relative gene expression was performed using the SYBR^®^
*Premix Ex Taq*^TM^ II (Tli RNaseH Plus) (TaKaRa, Shiga, Japan) kit in an Applied Biosystems StepOne^TM^ Plus Real-time PCR System (Thermo Fisher Scientific, Waltham, MA, USA). Thermal recycle regimes were as follows: 95 °C for 3 min, followed by 40 cycles of 95 °C for 10 s, then 60 °C for 30 s. A melting curve analysis was also conducted to ensure primer specificity of target genes: 95 °C for 15 s, 60 °C for 1 min, and 60–95 °C for 15 s (+0.3 °C per cycle) [85]. Expression data were normalized using the data of public reference genes *BnaEF1-α* [86] and *BnaGDI1* [87], and relative gene expression was calculated with the 2^−^^ΔΔC^*_T_* method [88]. Stem–loop reverse transcription PCR was used in the RT-qPCR experiments for miRNAs [89], and BnaU6 snRNA was used as an internal control in each reaction [90]. Sequences and amplification efficiency of primers used in this study are listed in additional file: Appendix A.

### 4.11. Statistical Tests

Significant differences (*, *p* < 0.05; **, *p* < 0.01; ***, *p* < 0.001) were determined by one-way analysis of variance (ANOVA), followed by Tukey’s honestly significant difference (HSD) multiple comparison tests, using the Statistical Product and Service Solutions 17.0 (SPSS, Chicago, IL, USA).

## Figures and Tables

**Figure 1 ijms-21-08453-f001:**
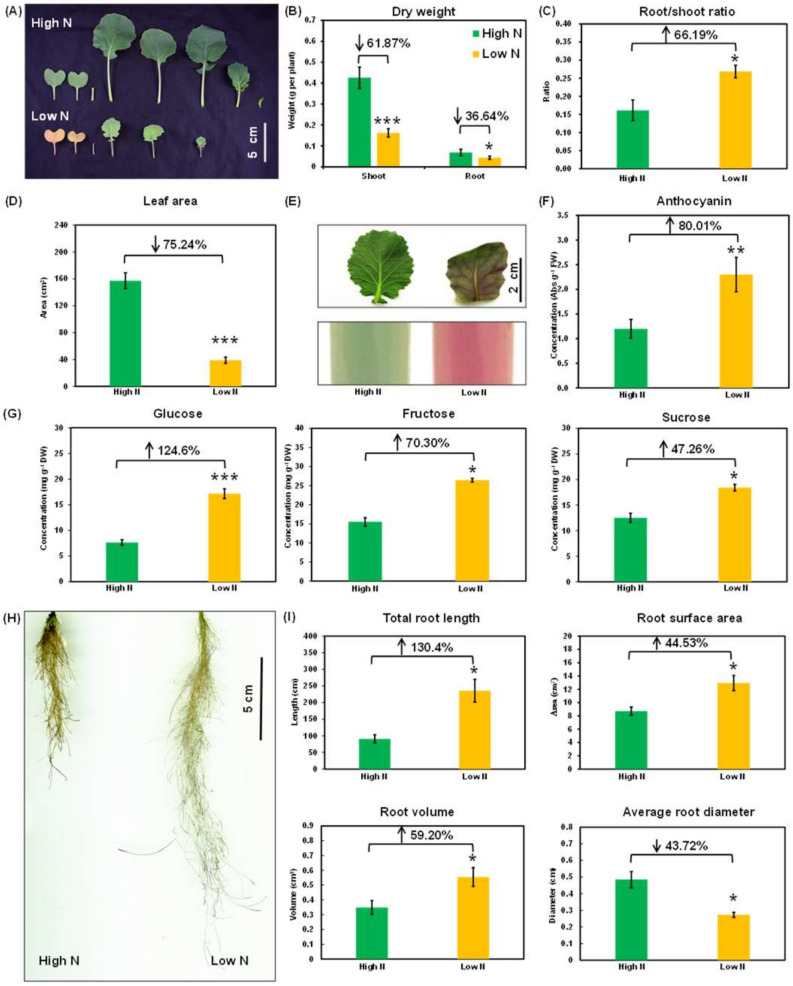
Morphological and physiological responses of rapeseed plants to nitrogen (N) limitations. (**A**) Leaf growth performance, (**B**) shoot and root dry weight, (**C**) ratio of root dry weight/shoot dry weight, (**D**) total leaf area, (**E**) leaf overaccumulating anthocyanin, (**F**) anthocyanin concentrations, (**G**) some carbohydrate (including glucose, fructose, and sucrose) concentrations in the leaves, (**H**) root growth performance, and (**I**) quantification of root system architecture of rapeseed plants that were hydroponically cultivated under N sufficiency and N limitation. Rapeseed plants after 7-day seed germination were hydroponically grown under high nitrate (6.0 mM NO_3_^−^) and low nitrate (0.3 mM NO_3_^−^) for 10 days, respectively. Columns denote means (*n* = 5), and error bars indicate standard deviations. *, *p* < 0.05; **, *p* < 0.01; ***, *p* < 0.001.

**Figure 2 ijms-21-08453-f002:**
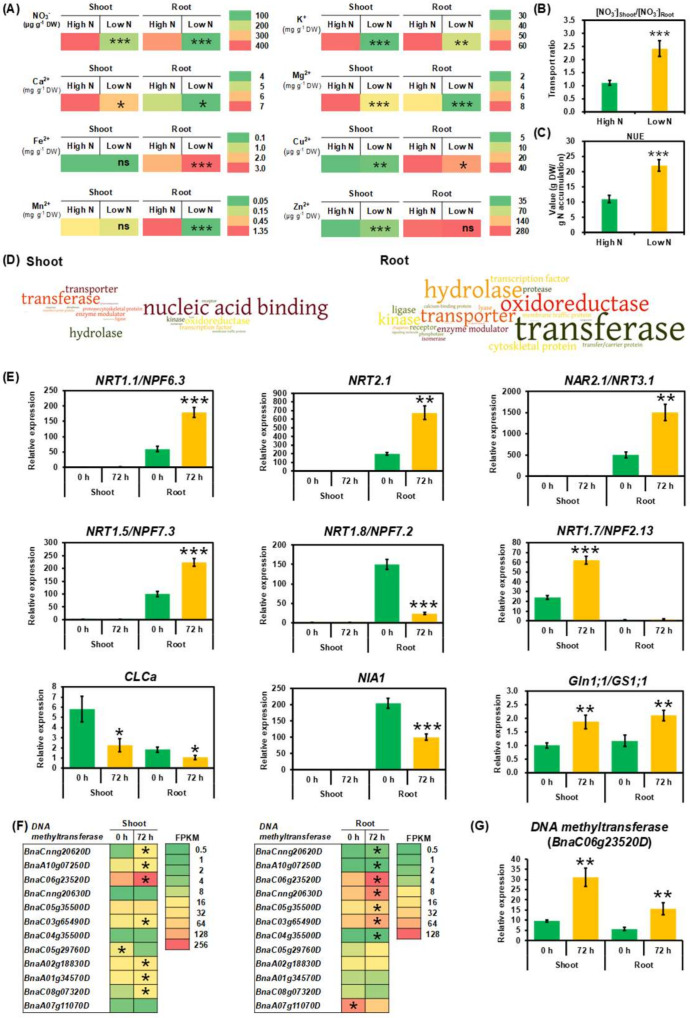
Ionomic and transcriptional responses of rapeseed plants to N limitation. (**A**) Ionomic profiling of rapeseed plants under N limitation, including the concentrations of nitrate (NO_3_^−^), potassium (K^+^), calcium (Ca^2+^), magnesium (Mg^2+^), iron (Fe^2+^), copper (Cu^2+^), manganese (Mn^2+^), and zinc (Zn^2+^) in the shoots and roots of rapeseed plants. (**B**) Ratio of nitrate concentrations in the shoots to those in the roots. (**C**) N use efficiency (NUE). NUE = total plant dry weight (g)/total plant N accumulation (g). (**D**) Gene Ontology (GO) enrichment analysis of the genes responsive to N limitation in the shoots and roots. The larger the font sizes, the more enriched the GO terms. (**E**) The qRT-PCR assay-assisted relative expression of some key N-metabolism-associated genes, including *NRT1.1/NPF6.3*, *NRT2.1*, *NAR2.1/NRT3.1*, *NRT1.5/NPF7.3*, *NRT1.8/NPF7.2*, *NRT1.7/NPF2.13*, *CLCa*, *Gln1;1/GS1;1*, and *NIA1*. (**F**) RNA-Seq assisted transcriptomic profiling of the differentially expressed DNA methyltransferase genes. Differentially expressed genes with higher expression between different treatments in the shoots or roots are indicated with asterisks. (**G**) RT-qPCR-assay-assisted relative expression of a DNA methyltransferase gene (BnaC06g23520D). For (**A**–**C**), rapeseed plants were hydroponically grown under high nitrate (6.0 mM NO_3_^−^) and low nitrate (0.3 mM NO_3_^−^) for 10 days until sampling, respectively. For (**D**–**G**), rapeseed plants were hydroponically grown under high nitrate (6.0 mM NO_3_^−^) for 10 days, and then they were transferred to high nitrate and low nitrate (0.3 mM NO_3_^−^) for 72 h until sampling, respectively. The times of 0 h and 72 h refer to the times after treatment of N deficiency. Columns denote means (*n* = 3), and error bars indicate standard deviations. *, *p* < 0.05; **, *p* < 0.01; ***, *p* < 0.001; ns, not significant.

**Figure 3 ijms-21-08453-f003:**
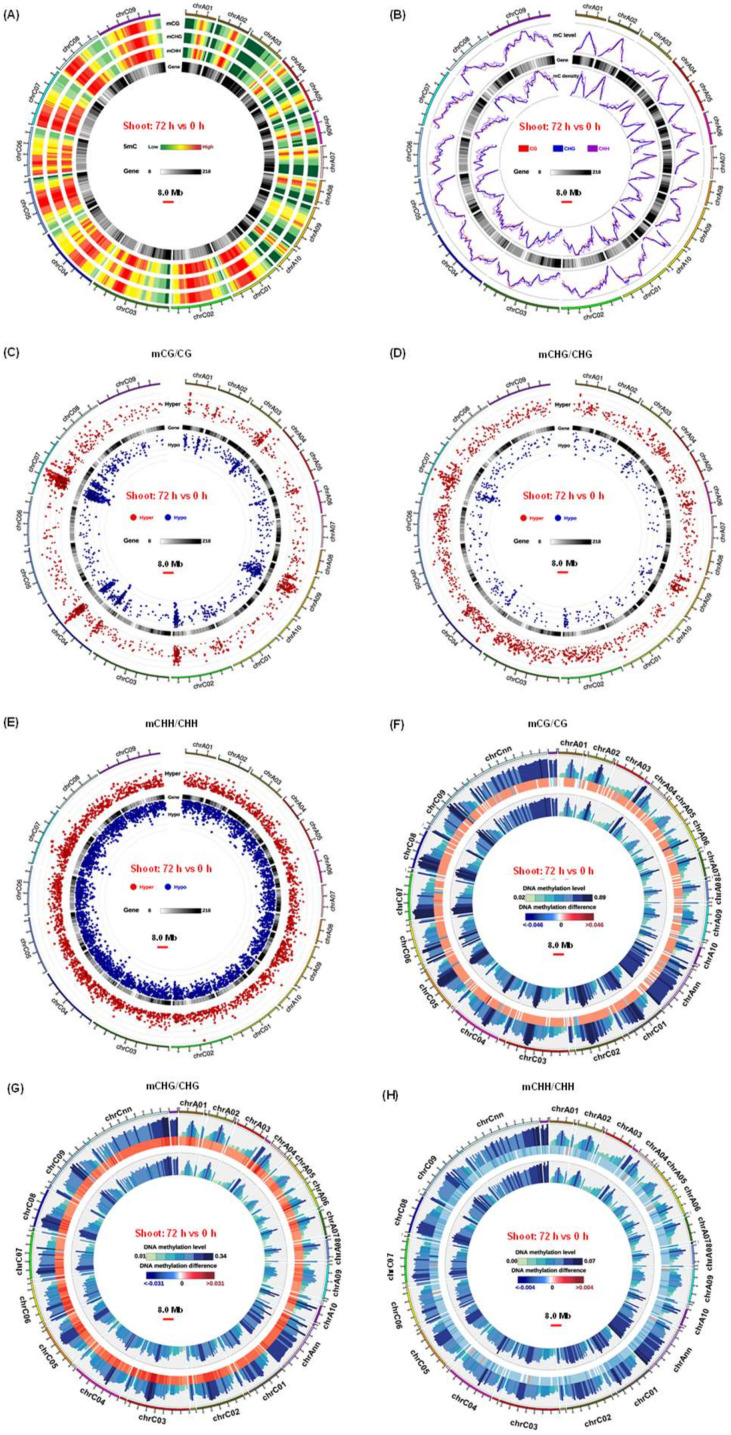
Overview of the genome-wide differential DNA methylation fingerprints in the rapeseed plant shoots between N sufficiency and N limitation. (**A**) Circos plots showing the density of differentially methylated regions across the rapeseed genome. In the Circos figure, the terms were as follows outside-to-inside: (i) chromosome, (ii) 5mC density in the CG, (iii) CHG, and (iv) CHH contexts, and (v) gene density of each chromosome in alloteraploid rapeseed. (**B**) Density and level plot of 5mC in CG, CHG, and CHH contexts in the gene bodies on each chromosome. In the Circos figure, the terms were as follows outside-to-inside: (i) chromosome, (ii) 5mC levels in the CG, CHG, and CHH contexts, (iii) gene density, and (iv) 5 mC densities in the CG, CHG, and CHH contexts in the gene bodies. (**C**–**E**) Density of hypermethylated/hypomethylated regions in the (**C**) CG, (**D**) CHG, and (**E**) CHH contexts across the rapeseed genome. In the Circos figure, the terms were as follows outside-to-inside: (i) chromosome, (ii) densities of hypermethylated regions in the CG, CHG, and CHH contexts, (iii) gene density, and (iv) densities of hypomethylated regions in the CG, CHG, and CHH contexts. (**F**–**H**) Differential methylation levels in the (**F**) CG, (**G**) CHG, and (**H**) CHH contexts across the rapeseed genome. In the Circos figure, the terms were as follows outside-to-inside: (i) chromosome, (ii) hypermethylation levels in the CG, CHG, and CHH contexts, (iii) DNA methylation differences, and (iv) hypermethylation levels in the CG, CHG, and CHH contexts. The times of 0 h and 72 h refer to the times after treatment of N deficiency.

**Figure 4 ijms-21-08453-f004:**
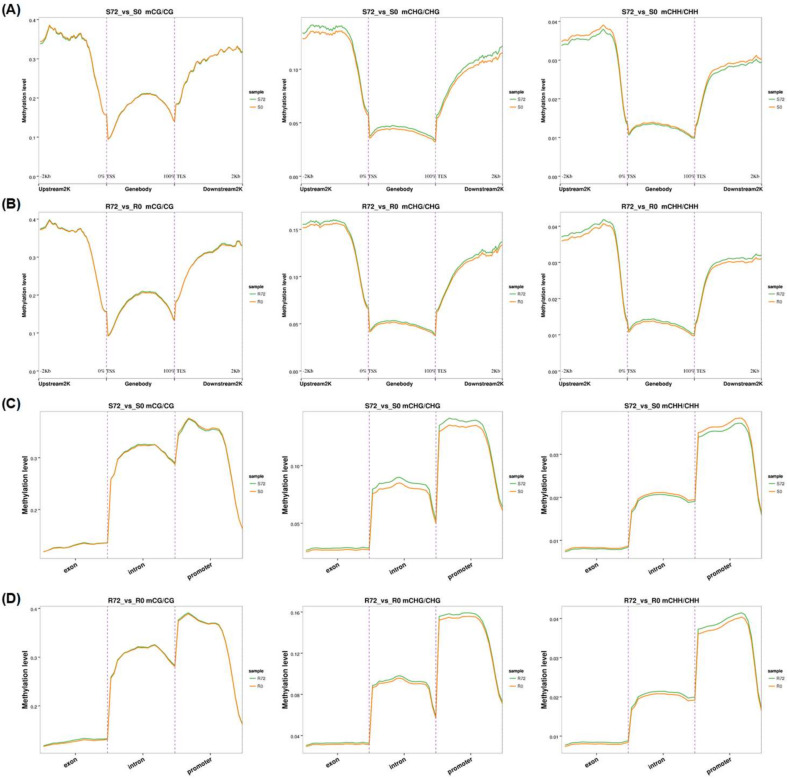
Comparison analysis of DNA methylation levels in different genomic functional regions in the shoots and roots of rapeseed plants under N sufficiency and N limitation. (**A**,**B**) DNA methylation levels in gene bodies and their flanking (2.0-kb upstream and downstream) regions under the CG, CHG, and CHH contexts in the (**A**) shoots and (**B**) roots. (**C**,**D**) DNA methylation levels in gene bodies (including exons and introns) and their promoter regions under the CG, CHG and CHH contexts in the shoots (**C**) and roots (**D**). Each functional region was equally divided into 20 bins, and the mean density of the methylated cytosines was defined as the methylation density in each bin. S, shoot; R, root. The times of 0 h and 72 h refer to the time after treatment of N deficiency.

**Figure 5 ijms-21-08453-f005:**
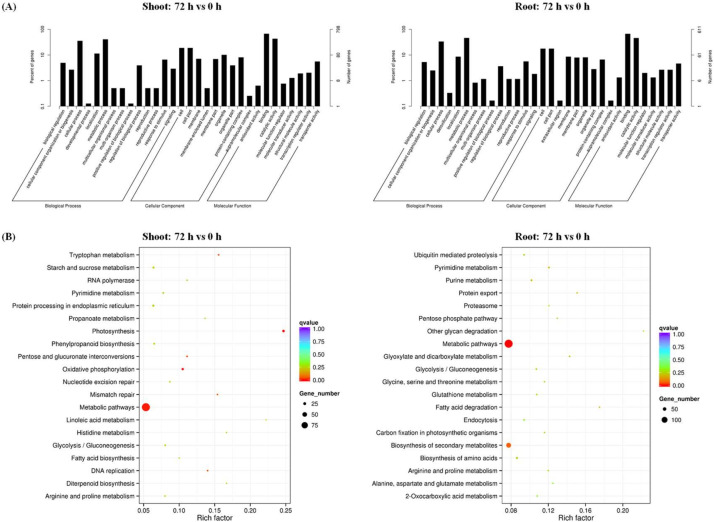
Functional enrichment analysis of differentially methylated genes (DMGs) between N sufficiency and N limitation in rapeseed plants. (**A**) Gene Ontology and (**B**) Kyoto Encyclopedia of Genes and Genomes (KEGG) pathway analysis of the DMGs in the shoots and roots of rapeseed plants under N limitation. The times of 0 h and 72 h refer to the time after treatment of N deficiency.

**Figure 6 ijms-21-08453-f006:**
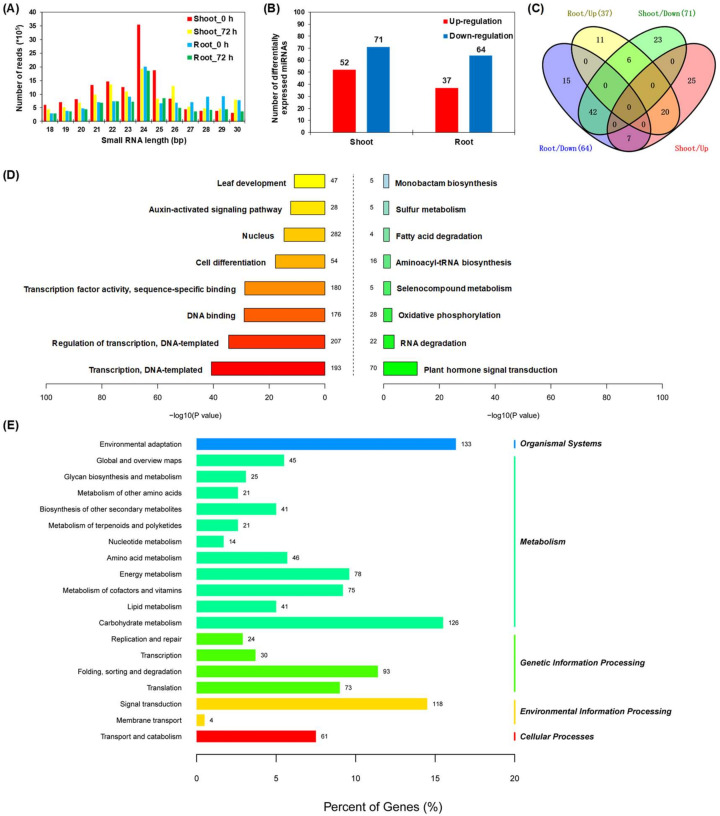
Molecular characterization of differentially expressed miRNAs in response to N limitation. (**A**) Length distribution of miRNAs in different sample libraries. (**B**) Number of differentially expressed miRNAs in the shoots and roots. (**C**) Venn diagram of differentially expressed miRNAs. Digit in each circle represented the number of differentially expressed miRNAs in the corresponding sample, and digit in the overlapping circle part represented the number of co-expressed miRNAs between different samples. (**D**) Gene Ontology and (**E**) KEGG pathway enrichment analysis of the genes targeted by differentially expressed miRNAs. In (**D**), *x*- and *y*-axis refers to the values of corrected −log_10_(*p*-value) and the number of genes targeted by differentially expressed miRNAs, respectively. In (**E**), *x*- and *y*-axis refers to the percentage of each category genes accounting for all the genes targeted by differentially expressed miRNAs and the target gene number. The times of 0 h and 72 h refer to the times after treatment of N deficiency.

**Figure 7 ijms-21-08453-f007:**
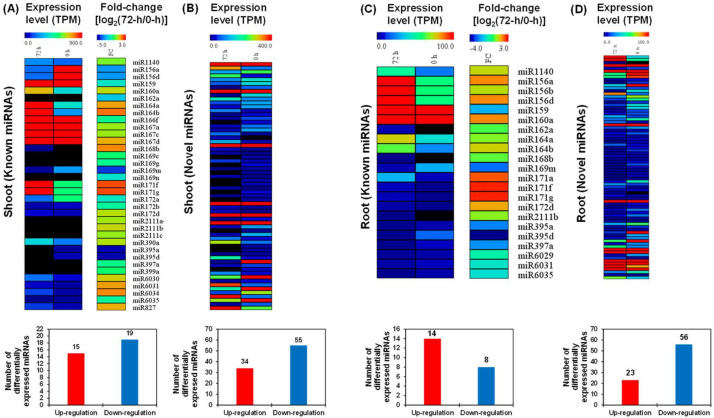
Heat-maps showing transcriptional profiling of differentially expressed miRNAs in response to N limitation. (**A**,**B**) Expression profiling, fold-change, and number of (**A**) known and (**B**) novel differentially expressed miRNAs in the shoots; (**C**,**D**) expression profiling, fold-change, and number of (**C**) known and (**D**) novel differentially expressed miRNAs in the roots. The times of 0 h and 72 h refer to the times after treatment of N deficiency.

**Figure 8 ijms-21-08453-f008:**
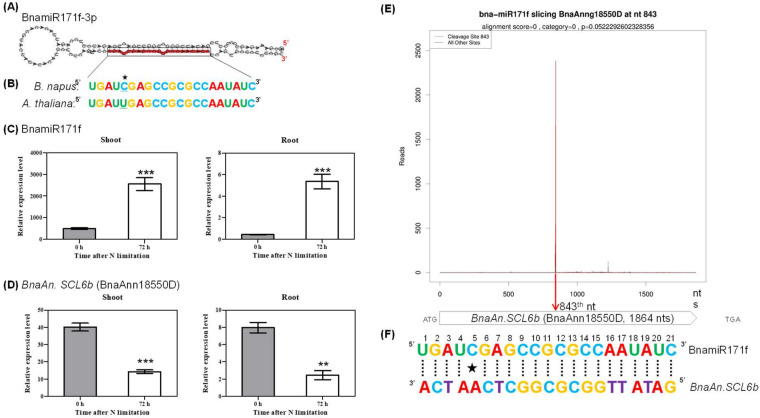
Molecular identification of the cleavage of *SCL6* by miR171 regulating root growth. (**A**) Hairpin structures and mature sequences of Bna-miR171f. The mature miRNA sequence was marked with red color. (**B**) Mature sequence alignment of Bna-miR171f between *Arabidopsis thaliana* and *B. napus*. The asterisk indicates the G:U pair. (**C**,**D**) Quantitative RT-PCR results showing relative expression levels of Bna-miR171f (**C**) and its target gene *BnaAn.SCL6b* (BnaAnn18550D) (**D**) in the shoots and roots under N sufficiency (0 h) and N limitation (72 h). Bars denote means (*n* = 3), and errors bars indicate standard deviation. Columns denote means (*n* = 3), and error bars indicate standard deviations. **, *p* < 0.01; ***, *p* < 0.001; ns, not significant. (**E**) Target plot (t-plots) of representative Bna-miRNA171 targets confirmed by degradome sequencing. The red lines showed the distribution of the degradome tags along the target mRNA sequences. The red arrow represented the cleavage nucleotide positions on the target gene. The *x*-axis indicates the nucleotide position in the target cDNA, and the *y*-axis indicates the number of reads of cleaved transcripts detected in the degradome cDNA library. Alignment score is the score for mismatch. Score = 0 represents perfect match and G:U = 0.5. (**F**) Sequence matching between the mature sequence of BnamiR171f and its target gene *BnaAn.SCL6b*. The asterisk indicates the G:U pair.

**Figure 9 ijms-21-08453-f009:**
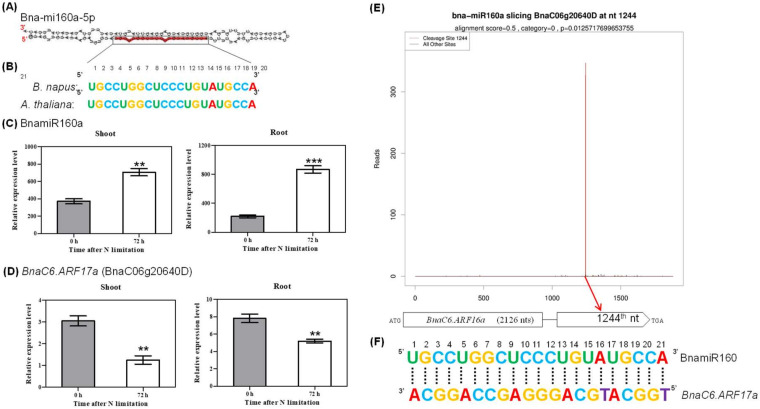
Molecular identification of the cleavage of *ARF17* by miR160 regulating root growth. (**A**) Hairpin structures and mature sequences of Bna-miR160a. The mature miRNA sequence was marked with red color. (**B**) Mature sequence alignment of BnamiR160a between *A. thaliana* and *B. napus*. (**C**,**D**) Quantitative RT-PCR results showing relative expression levels of (**C**) Bna-miR160a and (**D**) its target gene *BnaC6.ARF17a* (BnaC06g20640D) in the shoots and roots under N sufficiency (0 h) and N limitation (72 h). Bars denote means (*n* = 3), and errors bars indicate standard deviation. Columns denote means (*n* = 3), and error bars indicate standard deviations. **, *p* < 0.01; ***, *p* < 0.001; ns, not significant. (**E**) Target plot (t-plots) of representative Bna-miRNA160a targets confirmed by degradome sequencing. The red lines showed the distribution of the degradome tags along the target mRNA sequences. The red arrow represented the cleavage nucleotide positions on the target gene. The *x*-axis indicates the nucleotide position in the target cDNA, and the *y*-axis indicates the number of reads of cleaved transcripts detected in the degradome cDNA library. Alignment score is the score for mismatch. Score = 0 represents perfect match and G:U = 0.5. (**F**) Sequence matching between the mature sequence of BnamiR160a and its target gene *BnaC6.ARF17a*.

**Figure 10 ijms-21-08453-f010:**
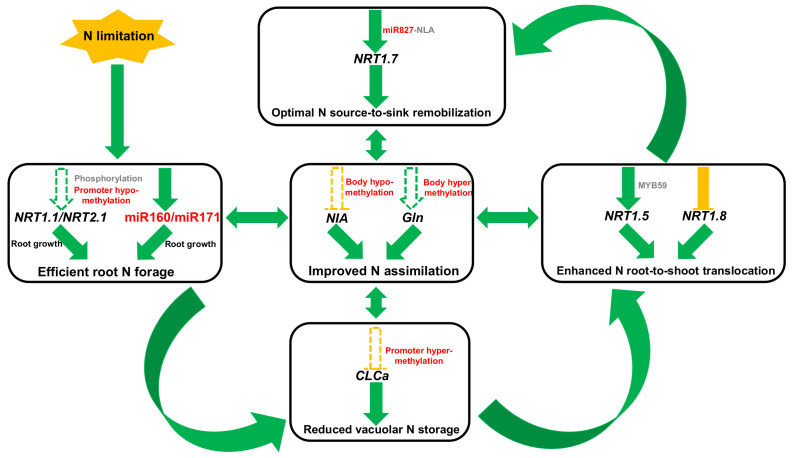
A proposed model showing the strategy of rapeseed plants to enhance N-limitation-triggered adaptation and use efficiency. Under N limitation, the phosphorylation and promoter hypomethylation resulted in the elevated expression of *NRT1.1* and *NRT2.1*, and the increased expression of miR160 and miR171, all of which contributed to stronger root system that facilitated efficient N forage. Limited N supply repressed the expression of *CLCa* through promoter hypermethylation regulation, and it reduced vacuolar N storage, which was favorable for N assimilation in the cytoplasm. In addition, N limitation also enhanced the *NRT1.5* expression and repressed the *NRT1.8* expression, both of which facilitated N translocation from roots to shoots. Furthermore, in the shoots, limitation of N induced an increase in the miR827 expression that downregulated the E3 ubiquitin ligase NLA, which accelerated the remobilization of N nutrients from old leaves to young ones through modulating *NRT1.7* degradation. The hypomethylation of the nitrate reductase *NIA1* and hypermethylation of *Gln1;1* within their gene bodies might be responsible for the reduced expression of *NIA1* and increased expression of *Gln1;1*. Green solid arrows and yellow solid lines indicate the valid processes or genes are induced and repressed by N limitation, and the dashed arrows indicate potential DNA methylation under N limitation. Red words indicate the epigenetic regulation of N-metabolism genes identified in this study, and grey words refer to regulatory mechanisms that were previously reported.

**Table 1 ijms-21-08453-t001:** Differential DNA methylation of some key N-metabolism-associated genes in the shoots and roots under N limitation.

Tissue	Chr	Start	End	Length	ML ^1^_72 h	ML_0 h	DM ^2^	C_Context	Gene ID	Region	Annotation
Shoot	chrC07	18318906	18318977	72	0.12	0.43	hypo	CG	BnaC07g12690D	promoter	*NRT1.6/NPF2.12*
	chrC07	18319032	18319214	183	0.03	0.10	hypo	CHH	BnaC07g12690D	promoter	*NRT1.6/NPF2.12*
	chrC06	31598782	31598876	95	0.23	0.13	hyper	CHH	BnaC06g30920D	promoter	*NRT1.7/ NPF2.13*
	chrCnn ^3^	57013740	57013864	125	0.24	0.36	hyper	CHH	BnaCnng57240D	promoter	*NRT1.11/NPF1.2*
	chrC04	32096889	32096944	56	0.34	0.10	hyper	CG	BnaC04g30250D	intron	*Gln1;1*
	chrC04	32096889	32096944	56	0.34	0.10	hyper	CG	BnaC04g30250D	exon	*Gln1;1*
	chrA05	17101604	17101779	176	0.14	0.05	hyper	CHH	BnaA05g22420D	promoter	*Gln1;3*
	chrC06	2820212	2820332	121	0.05	0.01	hyper	CHH	BnaC06g02010D	promoter	*Gln1;5*
Root	chrC08	19833588	19833747	160	0.14	0.23	hypo	CHH	BnaC08g15370D	promoter	*NRT1.1/NPF6.3*
	chrA06	2701052	2701177	126	0.04	0.11	hypo	CHH	BnaA06g04560D	promoter	*NRT2.1*
	chrC05	19042585	19042712	128	0.04	0.01	hyper	CHH	BnaC05g24580D	promoter	*NRT1.5/NPF7.3*
	chrC08	33965938	33966022	85	0.17	0.42	hypo	CHG	BnaC08g36990D	promoter	*NRT1.9/NPF2.9*
	chrCnn	14859936	14860033	98	0.10	0.04	hyper	CHH	BnaCnng15890D	promoter	*NAXT1/NPF2.7*
	chrA02	11371814	11371951	138	0.09	0.18	hypo	CHH	BnaA02g18610D	intron	*NIA1/NR*
	chrC04	32095812	32095894	83	0.34	0.06	hyper	CG	BnaC04g30250D	exon	*Gln1;1//GS1;1*
	chrA07	13217084	13217222	139	0.05	0.01	hyper	CHH	BnaA07g15180D	promoter	*CLCa*

Note: ^1^ ML, methylation level; ^2^ DM, differential methylation; ^3^ Cnn, refers to a certain chromosome that is unknown on the Cn subgenome; CLC, chloride channel; Gln, glutamine synthetase; NAXT, nitrate excretion transporter; NIA, nitrate reductase; NPF, nitrate/peptide family; NRT, nitrate transporter. The times of 0 h and 72 h indicate the time (hour) after the N limitation treatment.

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
