# Peer review of "Genome-Wide Differential DNA Methylation and miRNA Expression Profiling Reveals Epigenetic Regulatory Mechanisms Underlying Nitrogen-Limitation-Triggered Adaptation and Use Efficiency Enhancement in Allotetraploid Rapeseed"

_ijms, 2020, doi:10.3390/ijms21228453_

Round 1

Reviewer 1 Report

Hua et al. here report genome-wide epigenetic changes under N-starvation in oilseed rape (Brassica napus). B. napus seedlings were hydrophonically grown under the nutrient-controlled environment and changes in shoot and root morphologies were examined. In addition, global changes in transcriptome, DNA methylation and miRNA profiles were investigated under N-sufficient and –deficient conditions. Finally, the cause of low N-use efficiency (NUE) in allotetraploid B. napus was discussed at the gene regulation level. I think the authors performed comprehensive analysis to understand the effect of N-limitation on the growth of B. napus, while providing valuable information from the perspective of epigenetic regulation.

Some major comments are provided as below:

  1. The authors showed that expression levels of DNA methyltranserases were changed under N-limitation, which might lead to DMRs responsible for differential expression. Given the different roles of each class of DNA methylatransferase (DNMT), I would like the authors to classify their functions into de novo and maintenance DNMTs, and infer their contribution to changes in DNA methylation patterns according to the sequence contexts.
  2. In addition, I would like to suggest the authors to examine the possible roles of DNA demethylases responsible for active DNA demethylation because global loss of DNA methylation might result from the activation of DNA demethylases such as ROS1 homologs, rather than from downregulation of DNMTs.
  3. It would be more helpful if changes in DNA methylation at the representative loci were shown in more detail within the several-kb window, rather than at the genome-wide level. For instance, some candidate NRT genes showing differential expression (Figure 2E) should serve as good representatives.
  4. N-starvation appears to affect DNA methylation differently on subgenomes A and C (Figure 3), and the authors propose that differential TE compositions might be responsible for unique changes in DNA methylation profile to each subgenome (P21). If possible, it would be great if the authors could provide the effect of N starvation on the parental species B. rapa and B. oleracea, while comparing the differences between the parents and their hybrid B. napus in response to N-limitation.
  5. In Figure 10, the model is interesting but still very speculative and possibly misleading. More solid data should be required to support it. I would like to suggest the authors to generalize the model rather than to focus on specific genes and cases.

Minors:

P2 L87-88: The sentence reads a little awkward. Please re-write.

P17 last paragraph: play > plays

P19: Figure 3A > Figure 7A

P21: roleS > roles

Author Response

Response to Reviewer 1#

General comment: Hua et al. here report genome-wide epigenetic changes under N-starvation in oilseed rape (Brassica napus). B. napus seedlings were hydroponically grown under the nutrient-controlled environment and changes in shoot and root morphologies were examined. In addition, global changes in transcriptome, DNA methylation and miRNA profiles were investigated under N-sufficient and –deficient conditions. Finally, the cause of low N-use efficiency (NUE) in allotetraploid B. napus was discussed at the gene regulation level. I think the authors performed comprehensive analysis to understand the effect of N-limitation on the growth of B. napus, while providing valuable information from the perspective of epigenetic regulation.

Response: We appreciate for your warm work and thanks very much for your positive and constructive comments on our manuscript.

Your comments are considerably valuable and very helpful for revising and improving our paper, particularly on the experimental design for proceeding the project in the future. We have studied your comments carefully and have made corrections which we hope meet with your approval. The point-by-point responses to your comments are also listed as below.

Major concerns

Comment 1: The authors showed that expression levels of DNA methyltransferases were changed under N-limitation, which might lead to DMRs responsible for differential expression. Given the different roles of each class of DNA methyltransferase (DNMT), I would like the authors to classify their functions into de novo and maintenance DNMTs, and infer their contribution to changes in DNA methylation patterns according to the sequence contexts.

Response: Thank you very much.

As you have commented, DNA methylation of plant genomes is more extensive and affects a wider sequence diversity than in animals. It involves a larger set of specific DNA methyltransferases, some of which have no analogs in animals. Maintenance methylation of CG sites is carried out by DNA methyltransferase MET1 both during DNA replication and post-replication. A chromatin-remodeling protein DDM1 and three highly related SRA domain containing m5C-binding proteins, VIM1–VIM3, are most important assistants of MET1. A plant-specific chromomethylase CMT3 is major enzyme performing methylation of CHG sites, probably also in maintenance mode. A histone H3K9 methyltransferase KYP and CMT3 seem to be mutual helpers: CMT3 binds and methylates CHG sites in the H3K9/K27 methylated chromatin loci, whereas KYP binds and methylates histone H3 molecules in cytosine methylated loci. Asymmetric CHH sites cannot be methylated in a semi-conservative manner. Their methylation status in dividing cells is maintained by RNA-directed de novo DNA methylation (RdDM) activity of DRM methyltransferases with some help from CMT3. The DRM methyltransferases are probably responsible for de novo methylation of cytosine residues in all sequence contexts leading to creation of new methylated sites of all three types.

In this study, the differentially expressed DNA methylation transferases belong to the MET family members, which performed maintenance methylation of CG sites.

Comment 2: In addition, I would like to suggest the authors to examine the possible roles of DNA demethylases responsible for active DNA demethylation because global loss of DNA methylation might result from the activation of DNA demethylases such as ROS1 homologs, rather than from downregulation of DNMTs.

Response: Thank you very much for your kind suggestion.

Considering that DNA methyltransferase terms were highly enriched in the GO analysis, we decided to investigate differential expression profiling of DNA methyltransferase genes. We identified a total of 12 DNA methyltransferase genes that were differentially expressed in the shoots and roots under N limitation (Figure 2F). Among the eight DEGs in the shoots, most of them were upregulated, and similar expression pattern was also observed in the roots (Figure 2F). Subsequently, we selected a DNA methyltransferase gene (BnaC06g23520D) to examine its expression by RT-qPCR assays, whose result confirmed that N limitation led to a significant increase in the DNA methyltransferase expression (Figure 2G).

Indeed, we tested the transcriptional profiling of DNA demethylases under high N and low N conditions, however, we did not identify their differential expression. Therefore, we did not mention of the possible roles in the regulation of DNA methylation under N limitation in the Results part.

Comment 2: It would be more helpful if changes in DNA methylation at the representative loci were shown in more detail within the several-kb window, rather than at the genome-wide level. For instance, some candidate NRT genes showing differential expression (Figure 2E) should serve as good representatives.

Response: Thank you very much for your good advice.

In this study, as shown in Table 1, we have listed DNA methylation of some N-metabolism associated genes, which was as listed in the Results (Page 12) section of the manuscript.

  Among the genome-wide NRT1.1/NPF6.3 homologs and nine high-affinity NRT2.1 homologs in the rapeseed genome [28], we found only a CHH hypo-methylation site located in the promoter of an NRT1.1 homolog (BnaC08g15370D) and a CHH hypo-methylation site located in the promoter region of an NRT2.1 homolog (BnaC06g04560D) in the roots under N limitation, respectively (Table 1). N starvation-induced an increase in the expression of all the NRT1.1/NPF6.3 and NRT2.1 homologs (Figure 2E), which might be correlated with their hypo-methylation in their promoter regions. However, we also found that DNA methylation in promoter regions did not consistently repress gene expression levels. For instance, we identified a hyper-methylated site in the promoter of NRT1.5/NPF7.3 (BnaC05g24580D) in the roots and a hyper-methylated site in the promoter region of NRT1.7/NPF2.13 (BnaC06g30920D) in the shoots of rapeseed plants under N limitation (Table 1), whereas their expression was induced by N limitation (Figure 2E). Furthermore, some other NPF family genes, such as NRT1.6/NPF2.12, NRT1.9/NPF2.9, NRT1.11/NPF1.2 and NAXT1 (nitrate excretion transporter 1)/NPF2.7, also showed significantly differential methylation between N sufficiency and N limitation (Table 1). CLCa, a chloride channel gene, is mainly responsible for vacuolar nitrate influx [29], and we found a hyper-methylated site in its promoter region (Table 1), which was consistent with its reduced expression under N limitation (Figure 2E).

In addition to the molecular identification of NRTs, we also tested a hyper-methylated site in an intron of Nitrate Reductase 1 (NIA1, BnaA02g18610D) and a hyper-methylated site in an exon of Glutamate Synthetase 1;1 (Gln1;1, BnaC04g30250D) in the roots under N limitation (Table 1), which might be associated with their enhanced transcriptional expression [22]. These above findings indicated that N limitation had a significant effect on N-metabolism associated genes, including those responsible for N absorption, translocation, remobilization, and assimilation.

Table 1. Differential DNA methylation of some key N-metabolism associated genes in the shoots and roots under N limitation.

Tissue

Chr

Start

End

Length

ML1_72 h

ML_0 h

DM2

C_context

Gene ID

Region

Annotation

Shoot

chrC07

18318906

18318977

72

0.12

0.43

hypo

CG

BnaC07g12690D

promoter

NRT1.6/NPF2.12

chrC07

18319032

18319214

183

0.03

0.10

hypo

CHH

BnaC07g12690D

promoter

NRT1.6/NPF2.12

chrC06

31598782

31598876

95

0.23

0.13

hyper

CHH

BnaC06g30920D

promoter

NRT1.7/ NPF2.13

chrCnn3

57013740

57013864

125

0.24

0.36

hyper

CHH

BnaCnng57240D

promoter

NRT1.11/NPF1.2

chrC04

32096889

32096944

56

0.34

0.10

hyper

CG

BnaC04g30250D

intron

Gln1;1

chrC04

32096889

32096944

56

0.34

0.10

hyper

CG

BnaC04g30250D

exon

Gln1;1

chrA05

17101604

17101779

176

0.14

0.05

hyper

CHH

BnaA05g22420D

promoter

Gln1;3

chrC06

2820212

2820332

121

0.05

0.01

hyper

CHH

BnaC06g02010D

promoter

Gln1;5

Root

chrC08

19833588

19833747

160

0.14

0.23

hypo

CHH

BnaC08g15370D

promoter

NRT1.1/NPF6.3

chrA06

2701052

2701177

126

0.04

0.11

hypo

CHH

BnaA06g04560D

promoter

NRT2.1

chrC05

19042585

19042712

128

0.04

0.01

hyper

CHH

BnaC05g24580D

promoter

NRT1.5/NPF7.3

chrC08

33965938

33966022

85

0.17

0.42

hypo

CHG

BnaC08g36990D

promoter

NRT1.9/NPF2.9

chrCnn

14859936

14860033

98

0.10

0.04

hyper

CHH

BnaCnng15890D

promoter

NAXT1/NPF2.7

chrA02

11371814

11371951

138

0.09

0.18

hypo

CHH

BnaA02g18610D

intron

NIA1/NR

chrC04

32095812

32095894

83

0.34

0.06

hyper

CG

BnaC04g30250D

exon

Gln1;1//GS1;1

chrA07

13217084

13217222

139

0.05

0.01

hyper

CHH

BnaA07g15180D

promoter

CLCa

Note: 1ML, methylation level; 2DM, differential methylation; 3Cnn, refers to a certain chromosome that is unknown on the Cn sub-genome; CLC, chloride channel; Gln, glutamine synthetase; NAXT, nitrate excretion transporter; NIA, nitrate reductase; NPF, nitrate/peptide family; NRT, nitrate transporter.The time of 0 h and 72 h indicate the time (hour) after the N limitation treatment.

Comment 3: N-starvation appears to affect DNA methylation differently on subgenomes A and C (Figure 3), and the authors propose that differential TE compositions might be responsible for unique changes in DNA methylation profile to each subgenome (P21). If possible, it would be great if the authors could provide the effect of N starvation on the parental species B. rapa and B. oleracea, while comparing the differences between the parents and their hybrid B. napus in response to N-limitation.

Response: Thank you very much for your good advice.

It is worth noting that the mean methylation level of the An subgenome was significantly lower than that of the Cn subgenome under all contexts from both libraries. Chalhoub et al. observed that Cn genes presented 4% to 8% higher methylation than their homologous An genes from the methyl bisulfite sequencing of roots and leaves (Chalhoub et al., 2014). A similar result was obtained in cultured microspores by Li et al. (2016): the amount of methylated cytosines distributed on the Cn subgenome was obviously larger than that distributed on the An subgenome. It was hypothesized that the different DNA methylation levels between the An and Cn subgenomes can be attributed to the asymmetrical distribution of TE density in the genome of B. napus.

  According to your suggestion, in the near future, we will explore the effect of N starvation on the parental species B. rapa and B. oleracea, while comparing the differences between the parents and their hybrid B. napus in response to N-limitation.

References:

Chalhoub, B.; Denoeud, F.; Liu, S.; Parkin, I.A.; Tang, H.; Wang, X.; Chiquet, J.; Belcram, H.; Tong, C.; Samans, B.; et al. Early allopolyploid evolution in the post-Neolithic Brassica napus oilseed genome. Science 2014, 345, 950–953.

Li, J.; Huang, Q.; Sun, M.; Zhang, T.; Li, H.; Chen, B.; Xu, K.; Gao, G.; Li, F.; Yan, G.; et al. Global DNA methylation variations after short-term heat shock treatment in cultured microspores of Brassica napus cv. Topas. Sci. Rep. 2016, 6, 38401.

Comment 4: In Figure 10, the model is interesting but still very speculative and possibly misleading. More solid data should be required to support it. I would like to suggest the authors to generalize the model rather than to focus on specific genes and cases.

Response: Thank you very much.

As you have commented, the model in Figure 10 is still very speculative, and more solid data should be required to support it. To avoid misleading readers, we have redrawn the Figure 10 (Page 23) in the revised manuscript, which is also as listed below:

Figure 10. A proposed model showing the strategy of rapeseed plants to enhance N limitation-triggered adaptation and use efficiency. Under N limitation, the phosphorylation and promoter hypo-methylation resulted in the elevated expression of NRT1.1 and NRT2.1, and the increased expression of miR160 and miR171, all of which contributed to stronger root system that facilitated efficient N forage. Limited N supply repressed the expression of CLCa through promoter hyper-methylation regulation, and it reduced vacuolar N storage, which was favorable for N assimilation in the cytoplasm. In addition, N limitation also enhanced the NRT1.5 expression and repressed the NRT1.8 expression, both of which facilitated N translocation from roots to shoots. Further, in the shoots, limited N-induced increased the miR827 expression that down-regulated the E3 ubiquitin ligase NLA, which accelerated the remobilization of N nutrients from old leaves to young ones through modulating NRT1.7 degradation. The hypo-methylation of the nitrate reductase NIA1 and hyper-methylation of Gln1;1 within their gene bodies might be responsible for the reduced expression of NIA1 and increased expression of Gln1;1. Green solid arrows and yellow solid lines indicate the valid processes or genes are induced and repressed by N limitation, and the dashed arrows indicate potential DNA methylation under N limitation. Red words indicate the epigenetic regulation of N-metabolism genes identified in this study, and grey words refer to regulatory mechanisms that were previously reported.

Minor points

Comment 1: P2: L87-88: The sentence reads a little awkward. Please re-write.

Response: Thank you very much for your kind suggestion.

According to your advice, we have rewritten the sentence (Page 2) in the Results section of the revised manuscript, which is also as listed below:

Under N limitation, rapeseed plants showed smaller and chlorotic leaves (Figure 1A-D), which was also confirmed by lower biomasses (Figure 1B).

Comment 2: P17: last paragraph: play > plays

Response: Thank you very much.

According to your suggestion, we have corrected “play” into “plays” (Page 17) in the Results section of the revised manuscript, which is also as listed below:

In Arabidopsis, SCL6 (SCARECROW-Like 6) targeted by miR171 plays an important role in the regulation of shoot branch production and leaf morphogenesis [40].

Comment 3: P17: Figure 3A > Figure 7A

Response: Thank you very much.

According to your suggestion, we have corrected “Figure 3A” into “Figure 7A” (Page 17) in the Results section of the revised manuscript.

Comment 4: P21: roleS > roles

Response: Thank you very much.

According to your suggestion, we have corrected “roleS” into “roles” (Page 17) in the Discussion section of the revised manuscript, which is also as listed below:

Regulatory roles of DNA methylation in gene transcription are dependent on plant tissue types, methylated genomic contexts, and target gene regions [67].

Once again, special thanks for your kind suggestion and valuable comment.

Reviewer 2 Report

This article is very interesting and highlights a potential epigenetic mechanism of nitrogen limitation adaptation in rapeseed. Over all quality of the manuscript is very good, however, following suggestions would help to further improve the quality. Figure 4- Did authors showed both genes and transposable elements (TE) ? It would be clear if authors check the DNA methylation only on genes for this figure. They can still show DNA methylation on TE in supplementary. DNA methylation is normally more on TE than genes, so if they separate genes and TEs in this figure, then probably they would be able to see a clear difference in DNA methylation. Table 1- Authors shown the methylation levels in the promoter of certain gene. Difference in DNA methylation levels between 0 and 72h is sometime too small, we don’t know whether this is statistically significant or not. I would recommend doing statistical test because if they are not statistically different, then some of the conclusions authors drawn would be invalid. I would recommend authors to calculate DNA methylation levels in the whole promoter (lets say 1-1.5Kb) of these genes. Some minor corrections are; Figure 1 D, F, G, and I, please indicate Low and High N in the graph. Line 103- I don’t see the Fig 1L. Please either change the text or label in figure 1. What is ‘position’ on X axis in Figure S2? Please elaborate the figure legends. Line 88- please remove comma at the end.

Author Response

Response to Reviewer 2#

General comment: This article is very interesting and highlights a potential epigenetic mechanism of nitrogen limitation adaptation in rapeseed. Overall quality of the manuscript is very good, however, following suggestions would help to further improve the quality.

Response: We appreciate for your warm work and thanks very much for your positive and constructive comments on our manuscript.

Your comments are considerably valuable and very helpful for revising and improving our paper, particularly on the experimental design for proceeding the project in the future. We have studied your comments carefully and have made corrections which we hope meet with your approval. The point-by-point responses to your comments are also listed as below.

Major concerns

Comment 1: Figure 4- Did authors show both genes and transposable elements (TE)? It would be clear if authors check the DNA methylation only on genes for this figure. They can still show DNA methylation on TE in supplementary. DNA methylation is normally more on TE than genes, so if they separate genes and TEs in this figure, then probably they would be able to see a clear difference in DNA methylation.

Response: Thank you very much for your kind suggestion.

In this study, Figure 4 showed DNA methylation levels in gene bodies and their promoter regions under the CG, CHG and CHH contexts, and it did not contain the DNA methylation of transposable elements.

Comment 2: Table 1- Authors shown the methylation levels in the promoter of certain gene. Difference in DNA methylation levels between 0 and 72 h is sometimes too small, we don’t know whether this is statistically significant or not. I would recommend doing statistical test because if they are not statistically different, then some of the conclusions authors drawn would be invalid. I would recommend authors to calculate DNA methylation levels in the whole promoter (lets say 1-1.5Kb) of these genes.

Response: Thank you very much for your good advice.

In this study, each sample contained three independent biological replicates for genome-wide DNA methylation analysis. The methylation level of individual cytosines was calculated as the ratio of mC to total cytosines [mC/(mC+un-mC)]. Fold change at each site was calculated by comparing the methylation ratio under N deficiency to the methylation ratio under N sufficiency. As shown, the fold changes of methylation levels in the gene promoter regions were up to two or more, and significance levels have been tested through the Student-t test or one-way analysis of variance.

Comment 3: Figure 1 D, F, G, and I, please indicate Low and High N in the graph. Line 103- I don’t see the Fig 1L. Please either change the text or label in figure 1.

Response: Thank you very much for your kind suggestion.

According to your advice, we have labelled the “Low N” and “High N” in the Figure 1 of the revised manuscript, which is also as listed below:

In addition, we have corrected “Figure 1L” into “Figure 1I” (Page 4) in the Results of the revised manuscript.

Comment 4: What is ‘position’ on X axis in Figure S2? Please elaborate the figure legends.

Response: Thank you very much.

According to your suggestion, we have elaborated ‘position’ on X axis in Figure S2 of the revised manuscript, which is also as listed below:

Supplementary Figure S2 Genome-wide DNA methylation (CG, CHG and CHH) levels in the shoots and roots of rapeseed plants under both N sufficiency (0 h) and N limitation (72 h). The time of 0 h and 72 h indicate the time (hour) after N limitation treatment. The X axis indicates the nucletide postion of sequencing reads.

Comment 5: Line 88- please remove comma at the end.

Response: Thank you very much.

According to your suggestion, we have removed the comma at the end of Line 88 in the revised Results section.

Once again, special thanks for your kind suggestion and valuable comment.
